# Afázie u mluvčích českého znakového jazyka

## Pozadí

Afázie, tedy porucha jazykových schopností způsobená získaným poškozením mozku, je dlouhodobě zkoumána u mluvčích mluvených jazyků (např. Duffau et al., 2005; Liben & Jarema, 2006; Lehečková, 2016). Po zásadním zlomu, kdy Stokoe (1960) poukázal na to, že i znakové jazyky mají definiční rysy spojované s lidskými jazyky, a nejsou tedy pouhým souborem gest, vznikly v 80. letech první kazuistiky zaměřující se i na znakové jazyky (např. Poizner, Bellugi, Iragui, 1984). Ty poukázaly na to, že poškození levé hemisféry skutečně ovlivňuje produkci a percepci amerického znakového jazyka, zatímco produkce gest obvykle zůstala neporušena. V návaznosti na tyto kazuistiky poté vzniklo několik dalších studií, které došly k podobným závěrům i v dalších národních znakových jazycích (např. Corina, 1998; Pickell et al., 2005).

I přesto, že od prvních zjištění uplynulo již čtyřicet let, dosud bylo publikováno pouze několik studií věnujících se jazykovým poruchám ve znakových jazycích. Všechny studie se navíc zabývaly zejména produkcí a percepcí takových jazykových prostředků, které se vyskytují i v mluvených jazycích. Znakové jazyky však disponují i prostředky, které v řadě mluvených jazyků nenalezneme (např. simultánní konstrukce nebo využívání prostoru pro gramatické účely) a u kterých lze předpokládat, že budou nějakým způsobem ovlivněny poškozením jazykových oblastí v mozku.

Kromě výše uvedených skutečností navíc dosud chybí jakýkoli popis projevů afázie u českého znakového jazyka (ČZJ), což má praktické důsledky i mimo lingvistické odvětví, včetně absence komplexního přístupu k diagnostice a terapii fatických poruch u uživatelů ČZJ.

Tato studie se zaměřuje na specifické projevy afázie u mluvčích českého znakového jazyka a kladla si za cíl popsat jednotlivé reprezentace afázie napříč různými jazykovými rovinami, porovnat je s dosavadními poznatky o afázii v cizích znakových jazycích a prozkoumat konkrétní jazykové prostředky, které ve znakových jazycích dosud ani v zahraničí nebyly prozkoumány.

## Metodologie

Výzkumu se zúčastnili dva neslyšící probandi – rodilí mluvčí českého znakového jazyka se získaným poškozením mozku projevujícím se poruchami v oblasti produkce jazyka. Elicitace byla tvořena čtyřmi částmi:

1. Popis obrázkového příběhu „Frog, Where Are You?".

2. Polostrukturovaný rozhovor zaměřený na elicitaci konkrétních jazykových prostředků, jako jsou specifické znaky, inkorporace a záporné či tázací věty.

3. Porovnání aktuálního jazykového projevu pacientů s jejich videozáznamy před onemocněním, což umožnilo analyzovat změny v produkci znaků.

4. Komunikace pacientů s jejich rodinnými příslušníky na téma Vánoce, ve které byla elicitace zaměřena na produkci minulého a budoucího času; sekundárně se sledovala i

komunikace pacienta jako taková – interakce, upoutání pozornosti, náhradní komunikační strategie atd.

**Výsledky a závěr**

Analýza nasbíraného materiálu poukázala na to, že projevy afázie v ČZJ odpovídají zjištěním u jiných znakových jazyků. Na fonologické rovině byly například pozorovány záměny ve všech parametrech znaku, jako je tvar ruky, místo artikulace a pohyb (viz Obr. 1). Na morfologicko-syntaktické rovině bylo zaznamenáno chybné užívání shodových a prostorových sloves, včetně používání nevhodných klasifikátorů. Pacienti například použili sloveso v citátové formě nebo klasifikátor pro dvounohé bytosti při popisu čtyřnohých zvířat.

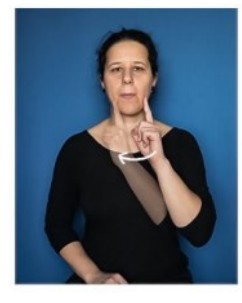
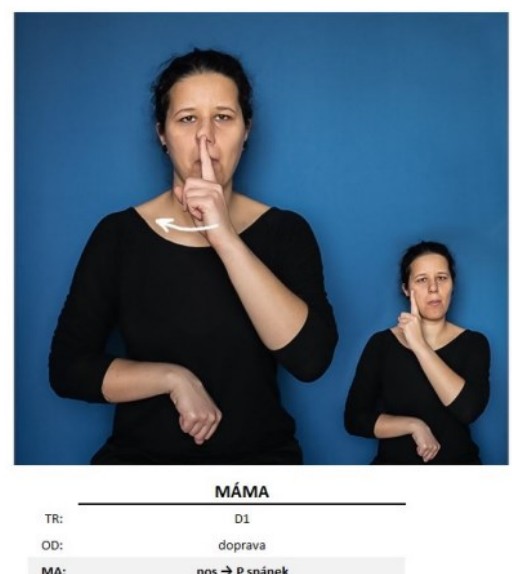

| MÁMA | |
|---|---|
| TR: | D1 |
| OD: | doprava |
| MA: | nos → P spánek |

**Obr. 1 – Substituce (místo artikulace)**
*pozn.: obrázek vlevo představuje správný znak, obrázek vpravo znak produkovaný probandem s afázií*

Kromě zmíněných skutečností se ukázalo, že poškození jazykových oblastí v mozku ovlivňuje i další jazykové prostředky, které dosud nebyly prozkoumány. Zaznamenáno bylo například chybné používání mimiky pro gramatické účely, čímž docházelo ke změnám významu celého sdělení, a to i přesto, že s nelingvistickým vyjádřením emocí pomocí výrazu obličeje neměli pacienti potíže. Afázie měla vliv i na produkci simultánních konstrukcí, kdy pacienti tyto konstrukce vyjadřovali sekvenčním způsobem (Obr. 2).

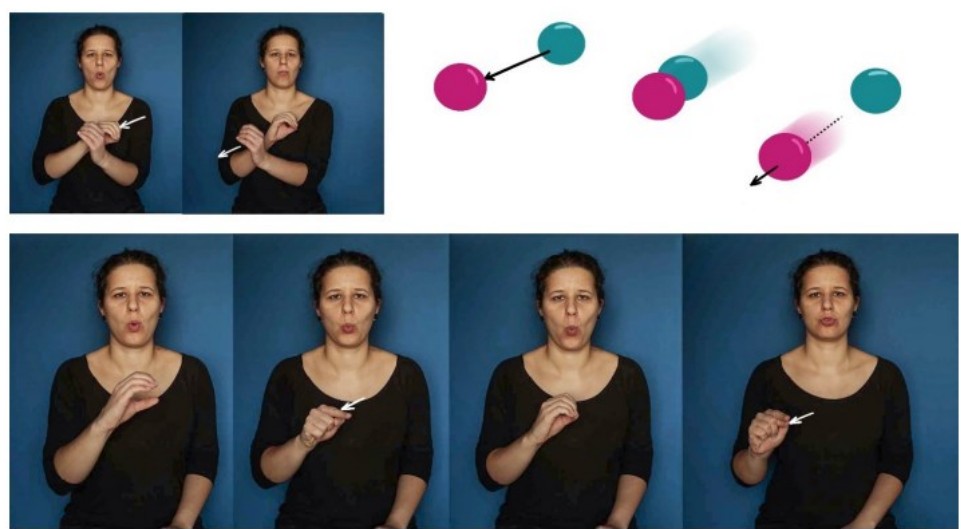

**Obr. 2 – Simultánní konstrukce**

Celkově byly afatické projevy u českého znakového jazyka podobné těm, které byly popsány u mluvených jazyků. Odlišnosti vycházely zejména z obecných rozdílů mezi znakovými a mluvenými jazyky, jako je jiná modalita obou jazyků (vizuomotorická vs. audioorální), strukturní odlišnost jazyků a výskyt některých jazykových prostředků, které v mluvených jazycích nenalezneme.

**Klíčová slova:** afázie, český znakový jazyk, neurolingvistika, jazykové poruchy, neslyšící

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
