# OpenReview forum: "Afázie u mluvčích českého znakového jazyka"
_CUNI.cz/2024/CJOLPhD — CUNI 2024 CJOLPhD Submission_

### Official Review · ~Anna_Staňková1 · 2025-01-07
**Dobrá práce!**

Abstrakt je dobře napsaný a strukturovaný. Abstrakt srozumitelně popisuje, co bylo cílem výzkumu, jaká byla metoda i jaké jsou výsledky. Na některých místech by se abstrakt dal trochu zkrátit, kdyby bylo potřeba dodržet nějaký limit slov/znaků. Myslím, že v úvodní části není potřeba vysvětlovat, co je afázie (to by měl recenzent - snad - vědět). Podobně bych si dokázala představit trochu úspornější popis předcházejícího výzkumu. Co se týče metodologie, je mi jasné, že pro takovou studii je velmi těžké sehnat potřebné participanty. Každopádně bych se pak v popisu výsledků vyvarovala silných závěrů jako "Kromě zmíněných skutečností se ukázalo, že poškození jazykových oblastí v mozku ovlivňuje i další jazykové prostředky, které dosud nebyly prozkoumány.", jelikož se výsledky opírají o data jen od dvou participantů (ale je to jen věc formulace).

---

### Official Review · ~Lucie_Jarůšková1 · 2025-01-08
**Pěkný abstrakt!**

Jedná se o velmi zajímavé téma, které je velmi potřebné zkoumat. Oceňuji vybrání klinické populace v českém znakovém jazyce, což je v nyní v daném oboru unikátní skupina a obecně srovnání projevů afázie u mluveného a znakového jazyka je chvályhodné.

Abstrakt je psán hezkou spisovnou češtinou, s jasnou strukturou. Nicméně si myslím, že je na konferenční poměry příliš rozsáhlý, za což může i použití odstavců či seznamů. Stálo by za to zkrátit především úvodní teoretickou část, která jde příliš do šířky a obecnosti. Působí spíše jako úvod článku (ačkoliv zajímavě prezentuje danou problematiku; především pro neznalce oboru poskytuje solidní základ).

Oceňuji přidání obrázků, především pro mluvčí mluveného jazyka slouží pro lepší představu. Závěry a výsledky bych interpretovala opatrněji z důvodu menšího počtu participantů.

Abstrakt mě velmi zaujal, výzkum by měl být rozhodně prezentovaný na nějaké konferenci a držím autorce palce s dalším bádáním.

---

### Official Review · ~Radek_Šimík1 · 2025-01-08
**Velmi zajímavý abstrakt, trošku moc dlouhé pozadí**

Pozadí je formulováno spíše jako úvod k článku než k abstraktu. Ideálně byste se měla "vejít" do délky, kterou má teď první odstavec. V závislosti na tom, na jakou konferenci byste abstrakt posílala, je například možné předpokládat znalost toho, co je to afázie, a není to tedy třeba vysvětlovat. Vypustit je též možné formulace typu "po zásadním zlomu...", navíc se opět jedná o něco, co je v široké lingvistické komunitě dobře známá věc, není to tedy třeba obhajovat nebo zmiňovat.

Zde jen stručný nástin toho, jak to může vypadat:

Poškožení levé periferie mozku může vést k afatickým projevům u mluvčích znakových jazyků (reference). Pro český znakový jazyk dosud výzkum o afatických projevech chybí, což má negativní důsledky ... Předložená studie si klade za cíl...

Obsahové doporučení: Asi by stálo za to získat i nějaká kontrolní data. Nejen od mluvčích před afázií (to je samozřejmě taky skvělé), ale na úplně stejných úlohách od mluvčích bez afázie.

Jinak výsledky a závěry jsou popsány velmi hezky. Bylo by skvělé, kdyby bylo možné výsledky aspoň bazálně kvantifikovat. Nyní není jasné, jestli se zmíněné obtíže projevily spíš izolovaně, nebo šlo o častý jev, a pak případně jak častý. (A i proto by bylo dobré mít data od kontrolní skupiny, aby bylo s čím srovnávat v těch specifických věcech.)

---

### Official Review · ~Maria_Onoeva1 · 2025-01-08
**Dobrá práce!**

Tento abstrakt představuje malou studii zaměřenou na uživatele českého znakového jazyka s afázií. Studie se účastnili pouze dva respondenti, přesto výsledky korespondují s dosavadními poznatky o mluvených jazycích a zároveň přinášejí nové informace specifické pro znakové jazyky.

Mezi silné stránky abstraktu patří jasně formulovaná výzkumná otázka, logická struktura textu a přehledné zpracování vizuálních prvků. Pro zlepšení by bylo vhodné více se zaměřit na význam dosažených výsledků a jejich možné dopady na další výzkum.

Otázky:
- Plánuješ zapojit víc účastníků? Pokud ano, změnila bys něco ve své metodě?

---

### Official Review · ~Barbora_Genserová1 · 2025-01-08
**Hezký abstrakt!**

Text je srozumitelně napsaný, logicky a přehledně strukturovaný, metodologie podrobně vysvětlená.
Na abstrakt je text příliš dlouhý. Přispívají k tomu obrázky, které bychom ale mohli brát podobně jako glosy a nepočítat do rozsahu abstraktu. Naopak dobře zkrátit by šel úvod (lze se spolehnout na to, že jsou obecné informace v lingvistické komunitě známé, a pokud nejsou, čtenáři si je dohledají podle citovaných referencí).
Pro bezpečnější závěry by bylo dobré provést studii s více participanty, i když je mi jasné, že sehnat takovouto skupinu bude náročné. Je ale skvělé, že jsi měla k dispozici srovnání jazykového projevu před afázií se současnou produkcí.